# Seasonally dependent increases in subweekly temperature variability over Southern Hemisphere landmasses detected in multiple reanalyses

**Patrick Martineau[1], Swadhin K. Behera[1], Masami Nonaka[1], Hisashi Nakamura[2,1], and Yu Kosaka[2]**

[1]Japan Agency for Marine-Earth Science and Technology, Yokohama, Japan
[2]Research Center for Advanced Science and Technology, The University of Tokyo, Tokyo, Japan

**Correspondence:** Patrick Martineau (pmartineau@jamstec.go.jp)

**Abstract.** The inter-dataset agreement of trends in subweekly near-surface (850 hPa) temperature variability over Southern Hemisphere midlatitude land masses is assessed among 12 global atmospheric reanalysis datasets. A comparison of the climatological temperature variance and dominant sources and sinks of the variance reveals that, except for NCEP-NCAR (R1) and NCEP-DOE (R2), there is a relatively good agreement for their magnitudes and spatial distributions during the satellite era (1980–2022), which indicates that the key features of subweekly variability are sufficiently well represented. A good agreement is noted for the positive trends found in subweekly variability over the satellite era affecting South Africa in September–October–November (SON) and South America in December–January–February (DJF). Although there is agreement in most of the reanalyses concerning the positive trend affecting Australia in SON, this has not yet emerged from the noise associated with interannual variability when considering only the satellite era. It is significant, however, when the period is extended (1954–2022) or limited to the most recent decades (1990–2022). The trends are explained primarily by a more efficient generation of subweekly temperature variance by horizontal temperature advection. This generation is also identified as a source of biases among the datasets. The trends are found to be reproduced even in those reanalyses that do not assimilate satellite data (JRA-55C) or that assimilate surface observations only (ERA-20C, 20CRv2c, and 20CRv3).

## 1 Introduction

Subweekly variability in the extratropics is produced by transient weather systems such as midlatitude cyclones/anticyclones, tropical cyclones migrating poleward, polar lows, and mesoscale storms exerting strong social impacts through the accompanying temperature and precipitation anomalies. Subweekly temperature variability, the focus of this work, is primarily generated by horizontal temperature advection. Amplification of temperature variance occurs when the advection of the climatological temperature gradient by subweekly wind anomalies acts to enhance subweekly temperature anomalies, i.e., when they induce fluxes of heat against the mean temperature gradients (Oort, 1964). This process describes the conversion of the available potential energy (APE) from the basic-state circulation to subweekly disturbances (or eddies), or in other words, the baroclinic conversion of energy. It is the dominant source of APE for eddies with periods shorter than 10 d (Sheng and Derome, 1991). Whereas horizontal motion generates temperature variance, vertical motion acts to dissipate it. Subweekly wind anomalies are generally upward where and when subweekly temperature anomalies are positive, counteracting the latter through adiabatic cooling to maintain thermal wind balance. The process primarily represents the conversion from APE to kinetic energy (KE) and is of a similar order of magnitude to baroclinic generation.

Trends in large-scale temperature gradients, brought about by human-induced radiative forcing, may alter the flow of energy between the mean state (mean APE) and transient eddies, and thus could potentially alter subweekly tem-

perature variability. Global warming simulations based on CMIP5 models project an amplification of subweekly temperature variability in the Southern Hemisphere (SH), which is mostly concentrated over the subpolar ocean ($\sim$ 55–60° S)
in December–January–February (DJF) but may impact land-masses such as South Africa and Australia in June–July–August (JJA) (Schneider et al., 2015). It is associated in part with an amplification of the meridional temperature gradient. Such amplification has been observed already in
extratropical cyclone activity (Reboita et al., 2015). Sub-weekly variability, as observed in the eddy KE, is also projected to amplify in CMIP6 models over the SH, but this increase is strongly underrepresented in contrast to three reanalysis datasets (Chemke et al., 2022). It is generally not
well known, however, how well subweekly temperature variability is represented in reanalyses and whether there is a good agreement concerning the trends observed in the past decades.

Discrepancies among reanalysis outputs may arise from
20 differences in the representation of sub-grid-scale physical processes among the forecast models, differences in their data assimilation system, and differences in the observations being assimilated (Fujiwara et al., 2017, 2022). It is well known that conventional observation data have been scarce in
the SH in contrast to the Northern Hemisphere (NH) (Noone et al., 2021), which can lead to comparatively larger uncertainties in the representation of atmospheric variability over the SH. Atmospheric circulation variability at the largest spatial scale, as captured by the annular mode indices (North-
ern Annular Mode in the NH and Southern Annular Mode in the SH), was shown to be more uncertain in the SH upper troposphere (Gerber and Martineau, 2018), especially before satellite observations became available for data assimilation. The agreement among the reanalysis datasets concern-
ing synoptic-scale subweekly variability near the surface was assessed in the context of extratropical storm tracks, with better agreement found in the NH compared to the SH (Wang et al., 2016). For example, Sang et al. (2022) found that inter-dataset differences in the representation of baroclinicity were
more pronounced in the SH than in the NH. Notably, in contrast to higher-resolution (newer) products, lower-resolution (older) products were found to underrepresent baroclinicity as well as eddy APE (i.e., 2–8 d temperature variance), especially in the upper troposphere. Their diagnostics, however,
were either shown as zonal averages, or vertically averaged quantities. The representation of the detailed spatial distributions of near-surface temperature variance and its trends in reanalyses remains largely unknown.

A comprehensive inter-comparison of the climatological
properties of SH subweekly temperature variability and its recent trends in 12 major global reanalysis datasets is thus carried out in this study. First, the climatological spatial distribution in the SH of near-surface (850 hPa) temperature variability and its dominant sources/sinks from 1980 to 2010
are investigated in a reanalysis ensemble mean (REM) of the

most recent reanalysis products, and the deviation of each reanalysis therefrom is also investigated. Then, the inter-reanalysis agreement in the trends is assessed with emphasis on midlatitude landmasses (South America, South Africa, and Australia), in recognition of the important socioeco-
60 nomic impacts associated with trends in subweekly temperature variance and the associated temperature extremes.

## 2 Methods

### 2.1 Reanalysis data

The reanalysis datasets used in this study are listed in Ta-
65 ble 1. They can be classified into three categories depending on the type of data assimilated. Full input reanalyses are the standard reanalyses that assimilate all available observations. Most of them span the satellite era starting in 1979 and onward, but some also provide data before (ERA5 in the form
of a back extension; JRA-55 and NCEP-NCAR (R1) as standard output). Surface input reanalyses assimilate only surface data and are typically used to investigate atmospheric variability over the past century, including long periods when neither satellite observations nor conventional radiosonde
observations were available. Finally, conventional-input reanalyses assimilate only conventional observations but not satellite measurements. JRA-55C is a conventional-input reanalysis that was produced to assess the impact of satellite data assimilation in contrast to JRA-55. Since ERA5, JRA-
55, and NCEP-NCAR (R1) do not assimilate satellite observations before 1979, they can be considered as conventional-input reanalyses before the satellite era. More details about which observations are assimilated by reanalysis datasets can be found in Fujiwara et al. (2017). Data sources for each re-
analysis are listed in Table 2.

To ensure fairness in our comparison and reduce computational costs, the reanalyses are first interpolated onto a 2.5° by 2.5° horizontal grid that matches that of the products provided on the coarsest grid (NCEP-NCAR (R1) and NCEP-
90 DOE (R2)). We note that it is the original model resolution of each product, not that of the interpolated data onto which we apply our diagnostics, that influences atmospheric variability at short timescales (Sang et al., 2022). Our analyses focus on the 850 hPa pressure level, which is close enough to the
95 surface but also sufficiently high to avoid missing data due to topography. Pressure level diagnostics are used to allow for an investigation of the processes responsible for temperature variability and its trends. Data at 925, 850, and 700 hPa are used to evaluate vertical derivatives. Variables analyzed
include temperature ($T$), meridional wind ($v$), zonal wind ($u$), and pressure velocity ($\omega$). Daily means are obtained by averaging four time steps that are common to all reanalysis datasets (00:00, 06:00, 12:00, and 18:00 UTC).

To assess whether the trends observed at 850 hPa in reanal-
105 yses are consistent with those observed at the surface, we in-

**Table 1.** Reanalysis datasets investigated.

| Name | Period | Assimilation | Reference |
|------|--------|-------------|-----------|
| 20CRv2c | 1948–2014 | Surface input | Compo et al. (2011) |
| 20CRv3 | 1948–2015 | Surface input | Slivinski et al. (2019) |
| CFSR/CFSv2[a] | 1979–2022 | Full input | Saha et al. (2010a, 2014) |
| ERA-Interim | 1979–2019 | Full input | Dee et al. (2011) |
| ERA5 | 1959–2022 | Full input | Hersbach et al. (2020) |
| ERA-20C | 1948–2010 | Surface input | Poli et al. (2016) |
| NCEP-NCAR (R1) | 1948–2022 | Full input | Kalnay et al. (1996) |
| NCEP-DOE (R2) | 1979–2022 | Full input | Kanamitsu et al. (2002) |
| JRA-55 | 1958–2022 | Full input | Kobayashi et al. (2015) |
| JRA-55C | 1958–2012 | Conventional input | Kobayashi et al. (2014) |
| MERRA[b] | 1979–2016 | Full input | Rienecker et al. (2011) |
| MERRA-2[b] | 1980–2022 | Full input | Gelaro et al. (2017) |

[a] CFSR/CFSv2c is obtained by merging CFSR and CFSv2c. We note that model resolution changed between the two and minor changes were made to parameterizations. [b] Only assimilated (ASM) products are used.

**Table 2.** Data source for each reanalysis.

| Dataset | URL/DOI | Last access |
|---------|---------|-------------|
| 20CRv2c | https://psl.noaa.gov/data/gridded/data.20thC_ReanV2c.html | 13 April 2020 |
| 20CRv3 | https://psl.noaa.gov/data/gridded/data.20thC_ReanV3.html | 12 May 2022 |
| CFSR/CFSv2 | https://doi.org/10.5065/D69K487J, https://doi.org/10.5065/D6N877VB | 5 December 2022 |
| ERA-Interim | https://www.ecmwf.int/en/forecasts/dataset/ecmwf-reanalysis-interim | 21 September 2017 |
| ERA5 | https://doi.org/10.24381/cds.bd0915c6 | 29 October 2022 |
| ERA-20C | https://doi.org/10.5065/D6VQ30QG | 31 December 2015 |
| NCEP-NCAR (R1) | http://www.esrl.noaa.gov/psd | 4 December 2022 |
| NCEP-DOE (R2) | http://www.esrl.noaa.gov/psd | 7 November 2022 |
| JRA-55 | https://doi.org/10.5065/D6HH6H41 | 1 May 2023 |
| JRA-55C | https://doi.org/10.5065/D67H1GNZ | 5 November 2017 |
| MERRA | https://doi.org/10.5067/8D4LU4390C4S | 4 October 2017 |
| MERRA-2 | https://doi.org/10.5067/QBZ6MG944HW0 | 22 November 2022 |

vestigate surface temperature data from the Berkeley Earth temperature record, a gridded station-based dataset (Rohde and Hausfather, 2020).

## 2.2 Subweekly temperature variability and its sources/sinks

By applying temporal filtering to the atmospheric thermodynamic equation to decompose temperature and wind variability into various frequency bands, one can obtain a budget for subweekly temperature variance ($\overline{T'^2}$ or $T_{\mathrm{VAR}}$) as TS1

$$\frac{\partial \overline{T'^2}}{\partial t} = \underbrace{-2\overline{T'\boldsymbol{v}'} \cdot \nabla \overline{T}}_{F_{\mathrm{horiz}}}$$

$$\underbrace{+2\overline{T'\omega'}\left(\frac{R\overline{T}}{c_p p} - \frac{\partial \overline{T}}{\partial p}\right)}_{F_{\mathrm{vert}}} + \chi, \tag{1}$$

where overbars denote the seasonal mean, and primes denote subweekly variability extracted with a 10 d high-pass filter. Here, $\chi$ represents forcing terms of comparatively lesser importance such as diabatic heating, cross-frequency interactions, and advection of $T_{\mathrm{VAR}}$ by the seasonal-mean circulation. When using reanalysis data, $\chi$ also includes the analysis increment, i.e., the correction performed during data assimilation, which may introduce an imbalance between the observed tendency and the generation/dissipation terms. The two leading forcing terms considered here include contributions from the horizontal advection of the seasonal-mean temperature by the horizontal subweekly wind component (first right-hand-side term; horizontal term or $F_{\mathrm{horiz}}$) and from the vertical advection of the seasonal-mean temperature and adiabatic expansion/compression by the vertical subweekly wind component (second right-hand-side term; vertical term or $F_{\mathrm{vert}}$). In Eq. (1), the temporally filtered thermodynamic equation is multiplied by $T'$ to obtain the tendency for temperature variance. As a consequence, $F_{\mathrm{horiz}}$ and

$F_{\text{vert}}$ are functions of horizontal and vertical fluxes of heat, respectively.

In the framework of atmospheric energetics (Lorenz, 1955; Oort, 1964), $F_{\text{horiz}}$ represents the APE conversion from the time-mean flow to subweekly eddies by horizontal winds. $F_{\text{vert}}$ represents both the conversion of eddy APE to eddy KE as well as the APE conversion from the seasonal-mean flow to subweekly eddies by vertical motions. The latter is in practice substantially smaller than the former and can be excluded from the energetics budget under scaling arguments (Tanaka et al., 2016). Thus $F_{\text{vert}}$ is considered here to primarily represent the conversion of eddy APE ($\sim T_{\text{VAR}}$) to eddy KE.

In this work, Eq. (1) is evaluated at 850 hPa to have sufficient spatial coverage above the earth's surface while still representing near-surface processes. It is assessed for each season (DJF, March–April–May (MAM), JJA, September–October–November (SON)) separately.

## 3 Results

### 3.1 Climatological properties of subweekly temperature variability

Climatological properties of subweekly temperature variability at 850 hPa ($T_{\text{VAR}}$) are first investigated for the period 1980–2010 for which all datasets are provided. They are assessed using the reanalysis ensemble mean (REM) which includes CFSR/CFSv2, ERA5, JRA-55, and MERRA-2, the current flagships from each reanalysis center (Fig. 1). $T_{\text{VAR}}$ is generally maximized at around 45° S over the South Atlantic Ocean and Indian Ocean in all seasons. This maximum is explained by the presence of the Antarctic polar frontal zone, a sharp gradient of sea surface temperature that anchors the midlatitude storm track (Nakamura et al., 2004; Nakamura and Shimpo, 2004), and accordingly, subweekly variability. Another prominent maximum in $T_{\text{VAR}}$ is observed over the southern Pacific at around 65° S. It exhibits a strong seasonality with a maximum in JJA and owes its existence to the amplified thermal contrasts at the sea–ice margin (Nakamura et al., 2004; Nakamura and Shimpo, 2004). Interestingly, secondary maxima are sometimes observed over or near landmasses in eastern South America, South Africa, and southern Australia. Their presence indicates that land–sea contrasts have the potential to anchor subweekly variability, like the Antarctic polar frontal zone. The South American maximum exhibits some seasonality, spreading over a greater land surface in JJA and SON, while being more concentrated and shifted to the south in DJF and MAM. The South African maximum tends to be stronger in SON and weakest in DJF and MAM. Of all three sectors, the Australian maximum shows the greatest seasonality with strongly amplified $T_{\text{VAR}}$ in SON and DJF and a clear minimum in JJA (Nakamura and Shimpo, 2004).

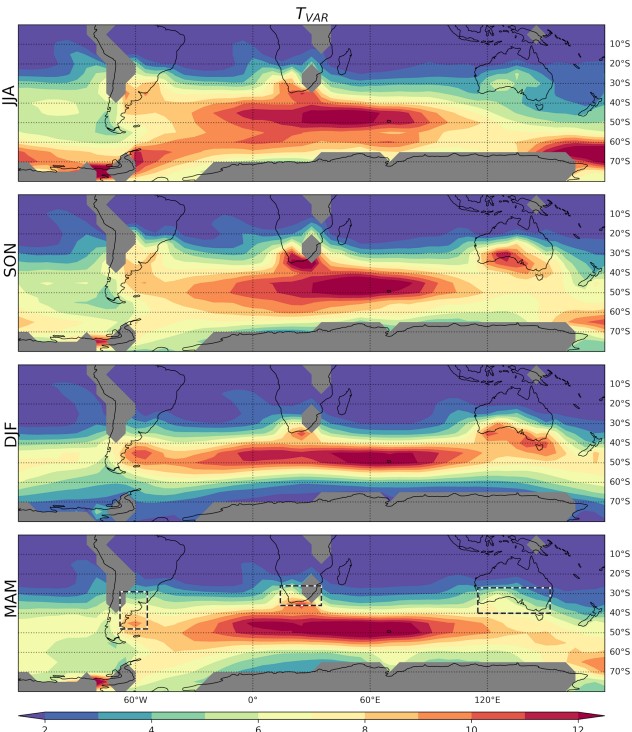

**Figure 1.** Climatology (1980–2010) of $T_{\text{VAR}}$ assessed at 850 hPa (shading; K$^2$) with the REM for the different seasons (rows). Areas below the earth's surface are masked in gray.

The spatial distribution and seasonality of $T_{\text{VAR}}$ correspond well to those of $F_{\text{horiz}}$ (Fig. 2). Its maxima are found in the midlatitude south Atlantic–Indian Ocean sector (year-round) and the subpolar south Pacific (especially in JJA) when and where the horizontal gradients of the climatological seasonal-mean temperature ($\nabla \cdot \overline{T}$; assessed with the spacing of $\overline{T}$ contours in Fig. 2) are stronger, providing favorable conditions for the baroclinic development of weather systems. Other maxima in $F_{\text{horiz}}$ and this gradient found over eastern South America, South Africa, and southern Australia exhibit the same seasonality as $T_{\text{VAR}}$, i.e., peaking in SON over South Africa and Australia and affecting a larger fraction of South American landmass in JJA and SON. These local maxima, which are comparatively greater than the gradient found over the oceans at similar latitudes, owe their existence to stationary waves associated with the distribution of oceans, landmasses, and topography (Wallace, 1983). These are also sectors where the correlation between $v'$ and $T'$ tends to be large and negative, indicating that the baroclinic structure of subweekly eddies is efficient in producing poleward fluxes of heat against the background temperature gradient (not shown).

As is evident in the right column of Fig. 2, $F_{\text{vert}}$ displays a similar spatial distribution to $F_{\text{horiz}}$ but of the opposite sign, contributing to dissipating $T_{\text{VAR}}$ over the vast majority of the SH. From an energetics perspective, it indi-

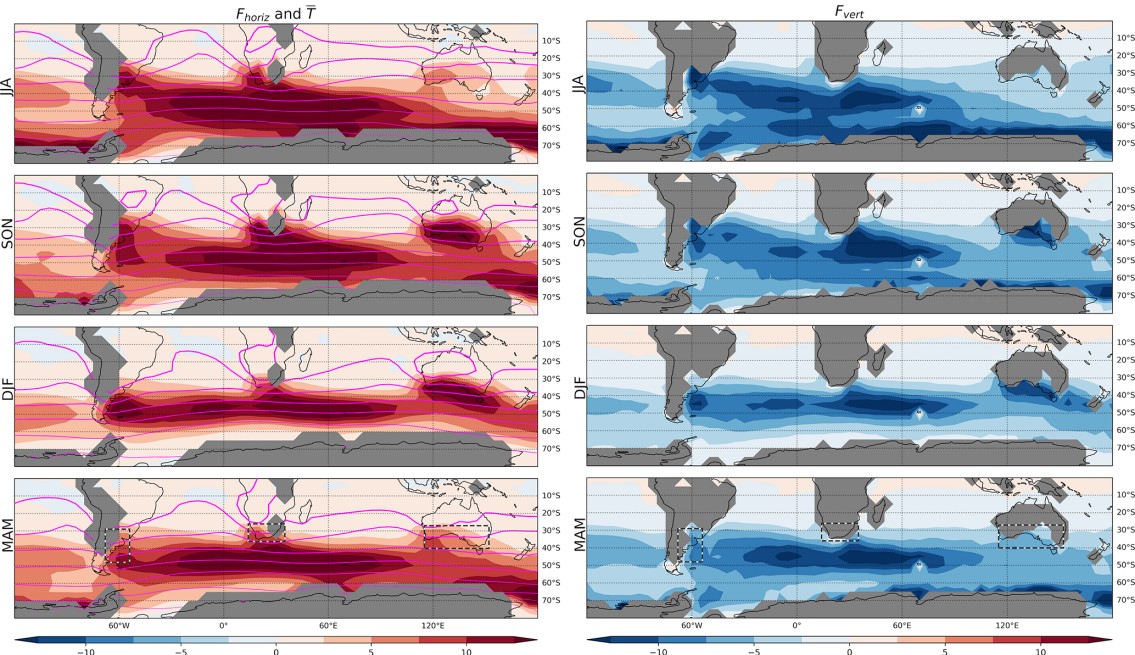

**Figure 2.** Same as in Fig. 1, but for (left) $F_{\text{horiz}}$ (shading; $\text{K}^2\,\text{d}^{-1}$) and (right) $F_{\text{vert}}$ (shading; $\text{K}^2\,\text{d}^{-1}$). The seasonal temperature climatology is overlaid on $F_{\text{horiz}}$ with purple contours at an interval of 5 K. Thicker contours indicate warmer temperatures.

cates the conversion from APE (temperature anomalies) to KE (wind anomalies) of subweekly eddies. The similarity between $F_{\text{horiz}}$ and $F_{\text{vert}}$ indicates that a significant fraction of eddy APE ($\sim T_{\text{VAR}}$) gained from the basic-state circulation by baroclinic energy conversion ($\sim F_{\text{horiz}}$) is immediately converted ($\sim F_{\text{vert}}$) to eddy KE. We note that $F_{\text{vert}}$ does not perfectly offset $F_{\text{horiz}}$, indicating that either other forcings or the analysis increments (both included in $\chi$ in Eq. 1) are not necessarily negligible. It is in fact known that diabatic processes, including heat exchanges with the underlying ocean (Nonaka et al., 2009), tend to dissipate temperature anomalies at that timescale.

Inter-reanalysis uncertainties in these basic properties of subweekly variability are then investigated further in SON, when $T_{\text{VAR}}$ is maximized in South Africa and southern Australia (Fig. 3). In general, there is a relatively good agreement for $T_{\text{VAR}}$ among the various reanalysis datasets. Even the surface-input reanalyses (20CRv2c, 20CRv3, ERA-20C), despite a deficit in the midlatitudes, overall capture the distribution of $T_{\text{VAR}}$. The modern full-input datasets tend to present only small biases relative to the REM climatology. Among all datasets, NCEP-NCAR (R1) and NCEP-DOE (R2) show the largest bias from the REM with negative biases reaching up to $\sim 2.7\,\text{K}^2$, which corresponds to up to $\sim 50\,\%$ of the REM climatology in some sectors. Whereas negative biases were found mostly over the ocean, weak positive biases were found over South Africa and southern Australia, which could be attributed to a greater density of observations available for assimilation. Comparing biases in the main gen-

eration term $F_{\text{horiz}}$ (Fig. 4) and $T_{\text{VAR}}$ (Fig. 3), we find a general correspondence between the two; biases in $T_{\text{VAR}}$ usually correspond to areas of same-signed biases in $F_{\text{horiz}}$. This is, however, not always the case. 20CRv2c, for instance, shows positive bias over the Indian Ocean, where $T_{\text{VAR}}$ is negatively biased. Biases in other forcing terms or compensation from the reanalysis increment (both included in $\chi$ in Eq. 1) may contribute to this mismatch. The large-scale features of these biases tend to be similar in other seasons (Figs. S1–S6 in the Supplement). For instance, the large negative biases affecting $T_{\text{VAR}}$ and $F_{\text{horiz}}$ in NCEP-NCAR (R1) and NCEP-DOE (R2) are present throughout the year.

## 3.2   Trends in subweekly temperature variability

In this section, we investigate trends in $T_{\text{VAR}}$ over the SH. We first focus on the period from 1980 to 2022 to assess the most recent trends during the satellite era. The trends are found to be spatially inhomogeneous with sectors of both decreasing and increasing $T_{\text{VAR}}$ (Fig. 5). When considering the entire SH, however, positive trends appear to dominate. This is especially true for the midlatitude storm track ($\sim 40$–$60°$ S). Over extratropical landmasses, we observe significant positive trends over midlatitude South America in DJF for which the reanalyses agree well. Positive trends are also observed in MAM, but the maximum is shifted southward ($\sim 50°$ S) and not as widespread and significant over land compared to DJF. Of all sectors, South Africa shows some of the largest positive trends in $T_{\text{VAR}}$ with significant positive trends in SON. While most reanalyses agree on positive trends in JJA,

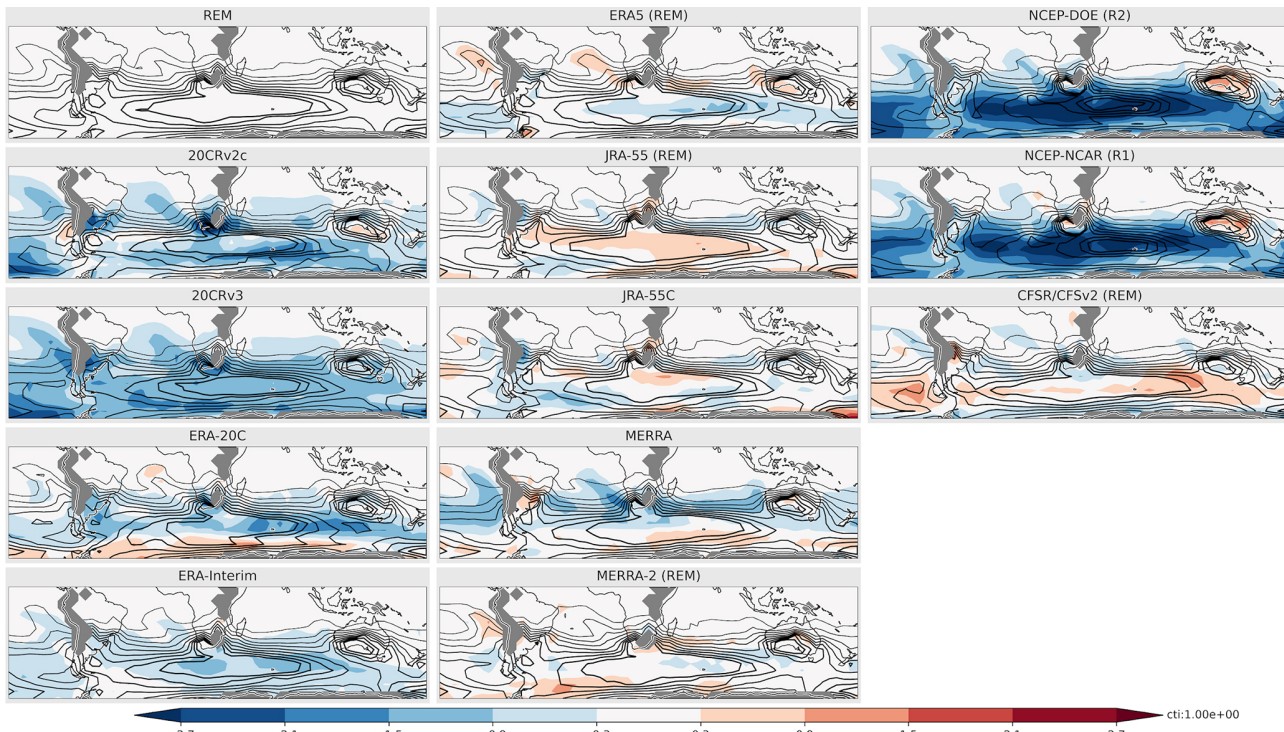

**Figure 3.** SON climatology (1980–2010) of $T_{VAR}$ (K$^2$; contour interval is indicated by "cti" next to the color bar) for the REM and individual reanalyses and biases from the REM (shading; K$^2$). The reanalyses included in the REM are labeled with (REM). Areas below the earth's surface are masked in gray.

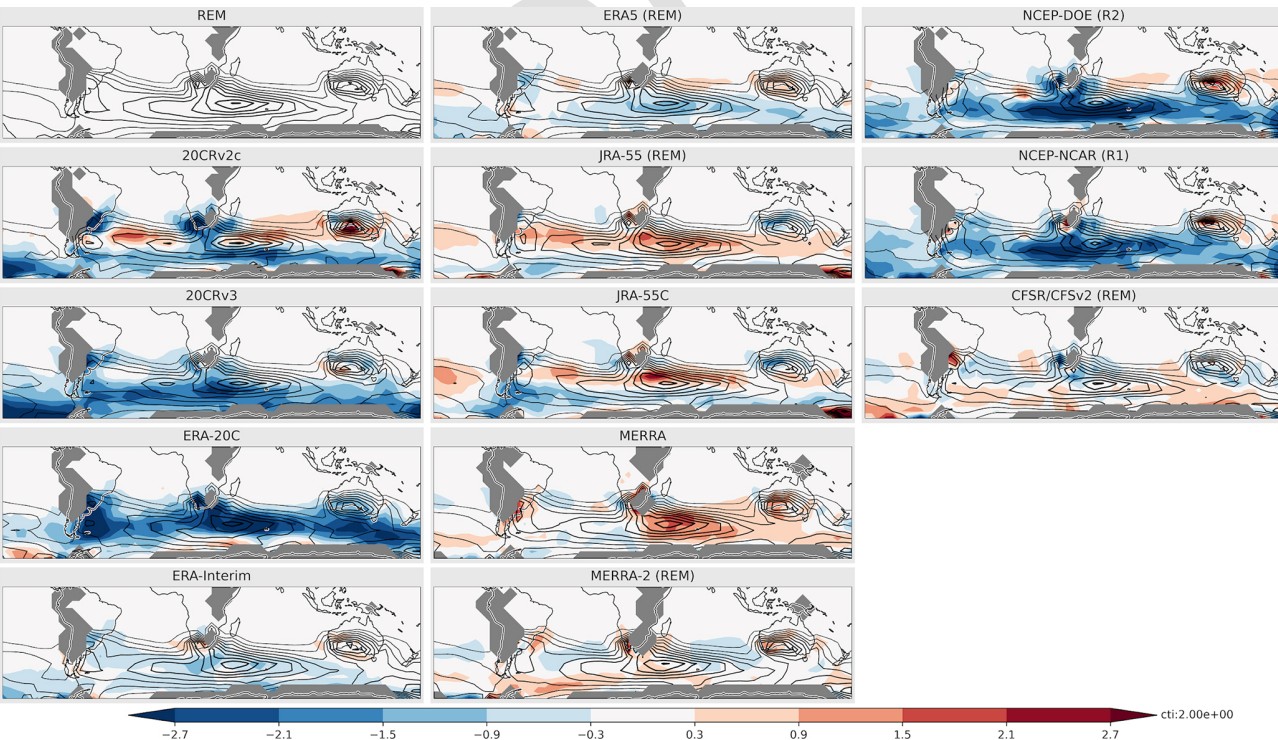

**Figure 4.** Same as in Fig. 3, but for $F_{horiz}$ (K$^2$ d$^{-1}$). The climatology is contoured at intervals of $2\,\mathrm{K}^2\,\mathrm{d}^{-1}$ with solid and dashed lines for positive and negative values, respectively. Thicker contours indicate larger magnitudes.

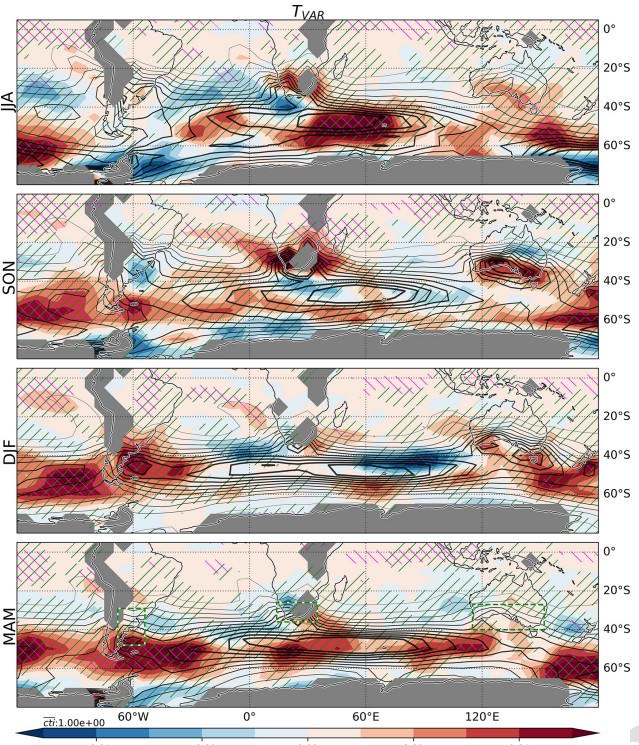

**Figure 5.** Trends of $T_{\mathrm{VAR}}$ (shading; $K^2\,yr^{-1}$) during 1980–2022 are shown for the REM for the different seasons (rows). The climatology is overlaid with contours at TS2 2 $K^2$ intervals. Thicker contours indicate larger magnitudes. Significant trends ($p < 0.05$) are indicated with purple hatching. Areas where more than one third of reanalyses agree on the sign of the trends are hatched in green.

they are not statistically significant. Although Australia is also found to be affected by positive trends in SON with a good agreement among the reanalyses, they are not statistically significant, either, for the period considered. Weaker trends are, however, observed in JJA over the southeastern Australian coast with more robust statistical evidence.

Most reanalyses agree concerning negative $T_{\mathrm{VAR}}$ trends affecting eastern South America in SON, South Africa in DJF and MAM, as well as northern Australia in SON, but only the trend in Australia is statistically significant in the REM. Some of the most robust negative trends in $T_{\mathrm{VAR}}$ are observed in DJF over the southern Indian Ocean, and in JJA over the south Pacific and south Atlantic, far from landmasses.

The evolution of $T_{\mathrm{VAR}}$ is investigated in more detail in Fig. 6 for the three major land sectors of interest. Despite the presence of time-mean biases in reanalyses as documented in the previous section, the year-to-year variability of $T_{\mathrm{VAR}}$ is relatively similar among the various datasets during 1980–2022 in all the sectors. Over South Africa, however, surface-input datasets such as 20CRv2c and to a lesser extent ERA-20C show weaker interannual variability and tend to be biased negatively, although we note an improvement in 20CRv3 over 20CRv2c. Over the other sectors, there

is marked agreement between full-input and surface-input datasets, indicating that surface observations alone are sufficient to constrain $T_{\mathrm{VAR}}$ over these sectors.

Trends in $T_{\mathrm{VAR}}$ are generally similar among the reanalysis datasets over the satellite era and tend to be consistent with the trends observed in station-based surface data (Berkeley Earth). Over South Africa, surface $T_{\mathrm{VAR}}$ trends have a greater signal-to-noise ratio than the 850 hPa $T_{\mathrm{VAR}}$ trends in the reanalyses and they are significant in JJA and DJF, seasons for which the reanalysis-based trends are not. SON $T_{\mathrm{VAR}}$ trends observed over Australia at the surface are also more obvious than those at 850 hPa. They are, however, not significant, most likely because they have not emerged yet from the large interannual variability. It is also important to mention that the positive trends observed over South America in DJF, and South Africa in SON appear to be stronger in the satellite era (1980–2022) compared to the prior decades. What appeared to be a positive trend affecting $T_{\mathrm{VAR}}$ over South America in SON before the satellite era has come to a halt afterward.

The sensitivity of $T_{\mathrm{VAR}}$ trends to the periods considered is confirmed in Fig. 7, which illustrates trends and their significance as computed for various periods. Many of the full-input reanalyses that extend back before the satellite era show negative trends during $\sim 1970$–1990 over South America (DJF) and South Africa (SON), as well as for $\sim 1960$–1978 over Australia (SON). The South American trends are, by contrast, positive when assessed for the $\sim 1954$–1980 period. Yet, it must be kept in mind that assessing trends over such short periods may capture apparent "inter-decadal variability" unrelated to climate change or discontinuities in assimilated observations, for example, at the beginning of satellite data assimilation in 1979 in full-input datasets. Discontinuities in assimilation, however, may not be the main factor here, since $T_{\mathrm{VAR}}$ in Berkeley Earth tends to show similar long-term tendencies. Figure 7 also reveals that trends affecting Australia are significant when assessing them for the whole period (1954–2022) or for the most recent decades (1990–2022), which shows the most rapid intensification in ERA5, JRA-55, and the REM (see also Fig. 6 for Australia in SON). We note that NCEP-NCAR (R1) shows more negative trends for South America in DJF during 1960–2022 compared to other reanalyses that provide extended data (Fig. 7). It appears to be linked with a negative $T_{\mathrm{VAR}}$ bias in the satellite era in contrast to the earlier period (Fig. 6). The corresponding negative trends are also observed, although to a lesser extent, in ERA5, but not in JRA-55. The negative trend in NCEP-NCAR (R1) is very similar to the surface $T_{\mathrm{VAR}}$ trends assessed in the Berkeley Earth dataset. Nevertheless, this does not mean that NCEP-NCAR (R1) is closer to reality in that sector compared to other reanalyses. It may be that it fails to adequately capture the differences in mechanisms driving surface and 850 hPa variabilities. Over other sectors, $T_{\mathrm{VAR}}$ trends in Berkeley Earth and reanalyses are qualitatively similar.

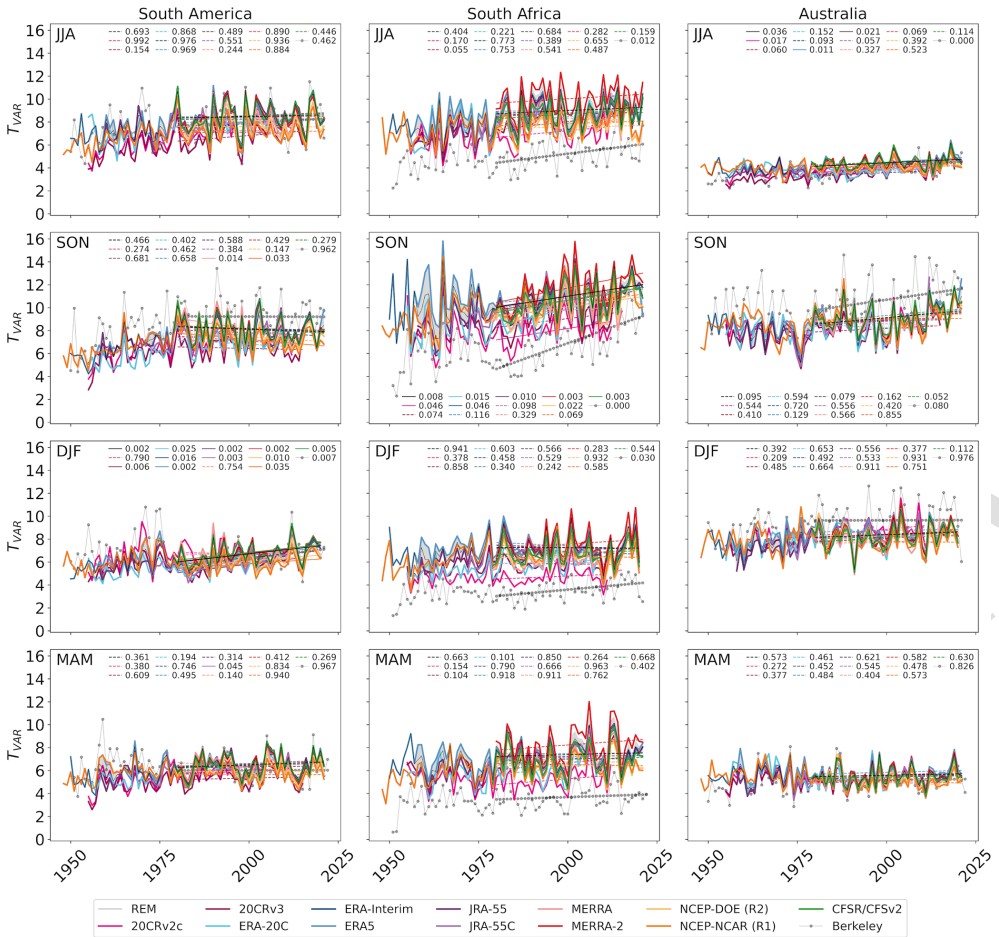

**Figure 6.** Time series of $T_{\mathrm{VAR}}$ ($K^2$) and its trend at three representative regions – South America (left), South Africa (middle), and Australia (right) – for different seasons (rows). The sectors over which $T_{\mathrm{VAR}}$ is averaged are illustrated with dashed boxes in the lower panels of Figs. 1, 2, and 5. Trends are computed for the period 1979–2022 (except for when datasets do not provide data for the full period) and illustrated with solid or dashed lines whether they are statistically significant or not (significant when $p < 0.05$). The $p$ value corresponding to each reanalysis is indicated in each panel. $T_{\mathrm{VAR}}$ from Berkeley Earth is assessed from observation-based data at the surface and scaled here by 2.5 for qualitative comparison with 850 hPa $T_{\mathrm{VAR}}$ in reanalyses.

We then turn our attention to the role of $F_{\mathrm{horiz}}$ in driving the observed $T_{\mathrm{VAR}}$ trends (Fig. 8). It is assessed by contrasting their spatial distributions (comparing Fig. 8 left column to Fig. 5). These two have similar distributions in the extratropics (pattern correlation of 0.62 for trends ranging from 80 to 20° S), confirming that the $T_{\mathrm{VAR}}$ trends primarily result from modulations of the baroclinic development of subweekly weather systems, i.e., changes in the associated heat fluxes against the background temperature gradient. Reanalyses agree about the prominent positive trends affecting southern Australia in SON, South Africa in SON and JJA, and midlatitude South America in DJF. However, the trends in $F_{\mathrm{horiz}}$ over landmasses are significant only over South Africa in SON for the period shown. Inspection of the meridional and zonal components of $F_{\mathrm{horiz}}$ (not shown) reveals that the trends over the SH are mainly contributed to

by trends in the meridional heat fluxes against the meridional gradient of seasonal-mean temperature ($-2\overline{v'T'}\frac{\partial \overline{T}}{\partial y}$).

One may consider that the $T_{\mathrm{VAR}}$ and $F_{\mathrm{horiz}}$ trends tend to exhibit good correspondence simply because they may both capture trends in subweekly eddy amplitudes. For instance, eddies of the same structure, if of larger amplitude, will yield both larger $T_{\mathrm{VAR}}$ and $F_{\mathrm{horiz}}$. This example illustrates that $F_{\mathrm{horiz}}$ is inadequate to identify the source of the amplified $T_{\mathrm{VAR}}$. To factor out the impact of eddy amplitude from $F_{\mathrm{horiz}}$ and thereby obtain an appropriate measure of $T_{\mathrm{VAR}}$ generation efficiency, we here divide $F_{\mathrm{horiz}}$ by the square root of the product of local eddy wind and temperature variance. For the meridional component of $F_{\mathrm{horiz}}$, this efficiency $\left(F_y^{\mathrm{eff}}\right)$ takes the form $-2\left(\frac{\overline{T'v'}}{\sqrt{\overline{T'^2}\,\overline{v'^2}}}\right)\frac{\partial \overline{T}}{\partial y}$, which is essentially the product of the local correlation between $T'$ and $v'$ and the

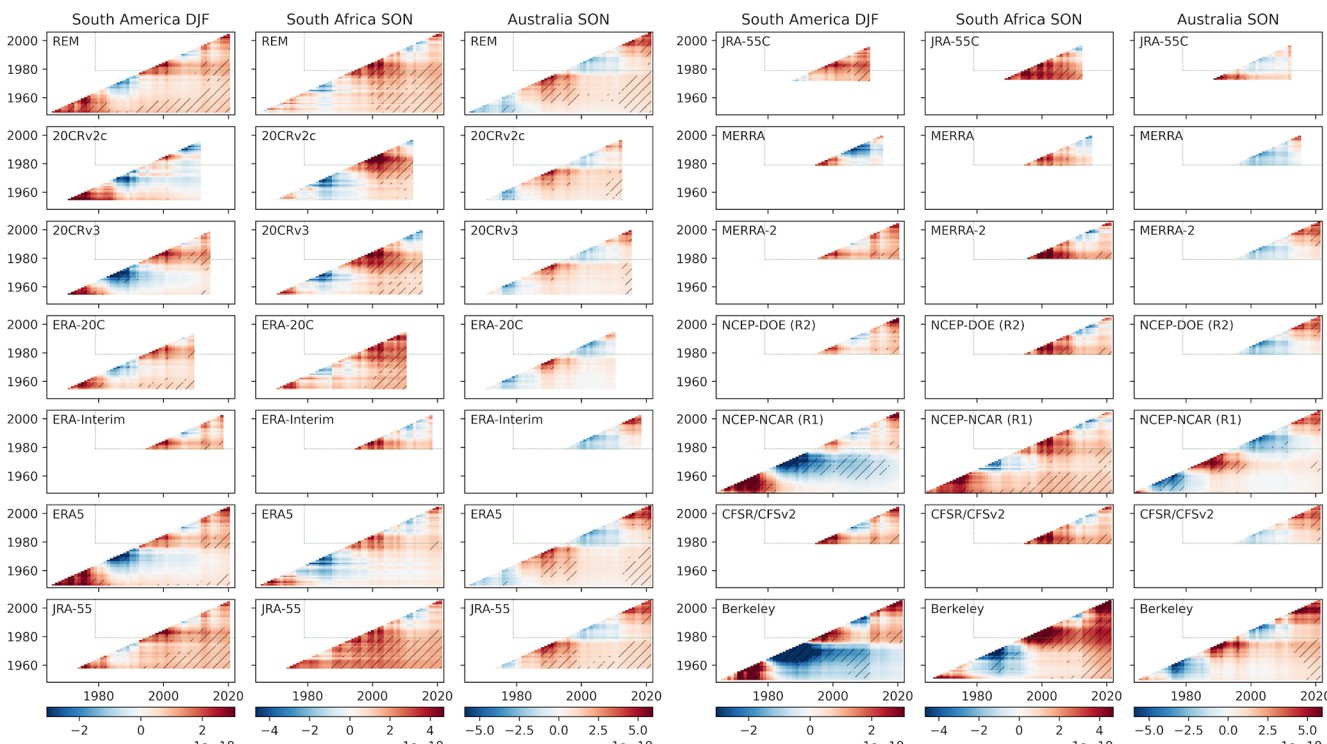

**Figure 7.** The sensitivity of trends in $T_{\mathrm{VAR}}$ ($\mathrm{K}^2\,\mathrm{yr}^{-1}$) to the period sampled is assessed over South America in DJF (left), South Africa in SON (middle), and Australia in SON (right). The sectors over which $T_{\mathrm{VAR}}$ is averaged are illustrated with dashed boxes in the lower panels of Figs. 1–2, 5, and 8–9. Significant trends ($p < 0.05$) are hatched in black. Trends assessed within the satellite era are delimited by dashed green lines. The $y$ and $x$ axes indicate the beginning and end, respectively, of the periods over which trends are assessed. $T_{\mathrm{VAR}}$ from Berkeley Earth is assessed from observation-based data at the surface and scaled here by 2.5 for qualitative comparison with 850 hPa $T_{\mathrm{VAR}}$ trends in reanalyses.

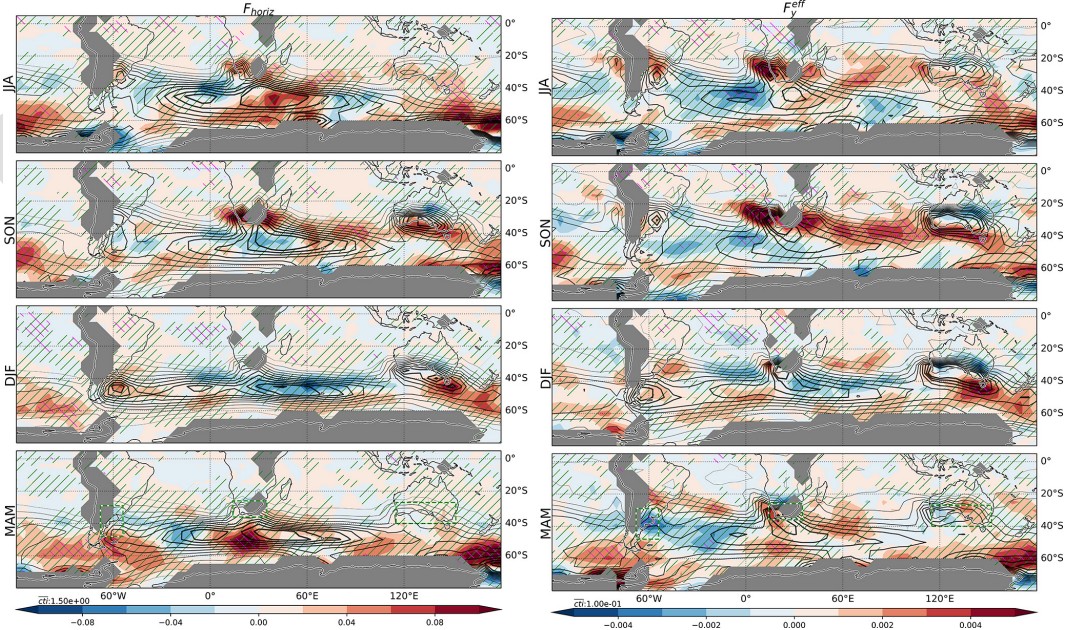

**Figure 8.** Same as in Fig. 5, but for (left) $F_{\mathrm{horiz}}$ ($\mathrm{K}^2\,\mathrm{d}^{-1}\,\mathrm{yr}^{-1}$; shading) and (right) $F_y^{\mathrm{eff}}$ ($\mathrm{K}\,\mathrm{m}^{-1}\,\mathrm{yr}^{-1}$; shading). The contour intervals of the climatology are indicated by "cti" just above the color bars.

meridional temperature gradient in the background state. The trends in the efficiency thus defined (Fig. 8, right column) exhibit qualitatively similar spatial distribution to the corresponding trends in $F_{\mathrm{horiz}}$ and thus explain well the $T_{\mathrm{VAR}}$
trends. We note that, when expressed as efficiency, trends in $F_{\mathrm{horiz}}$ become significant over Australia in SON and in the midlatitude south Indian Ocean. This enhanced generation efficiency can contribute to the Australian $T_{\mathrm{VAR}}$ trends through the upstream generation of subweekly disturbances
and the subsequent advection of $T_{\mathrm{VAR}}$ by the westerly winds.

In the extratropics, positive trends in $F_{\mathrm{horiz}}$ efficiency are generally collocated with trends in the magnitude of the climatological temperature gradient (Fig. 9, left column). Most of these changes are explained by trends in the meridional
temperature gradient ([TS3]$|\partial \overline{T}/\partial y|$, not shown). Amplified gradients are notably observed along the southern coast of Australia in SON, and South Africa in JJA and SON. In South America, by contrast, the correspondence between the trends in $\left|\nabla \overline{T}\right|$ and $F_{\mathrm{horiz}}$ is not clear. For instance, the tem-
perature gradient in DJF is found to weaken over sectors of positive $F_{\mathrm{horiz}}$ trends. We find that over that sector, the amplifying generation is attributable to the more favorable structure of baroclinic growth of subweekly anomalies. The correlation between $-v'$ and $T'$ shows positive trends (red shading
in Fig. 9, right column). Since their correlation is typically positive over that sector (poleward eddy heat fluxes), it represents an increase in the efficiency of subweekly eddies to produce heat fluxes against the Equator-to-pole temperature contrast. Trends in $F_{\mathrm{horiz}}$ over South Africa and Australia, by
contrast, are dominated by the strengthening of the meridional temperature gradient, and only weak trends in the correlation between $-v'$ and $T'$ are observed over these sectors. We note, however, that just west of South Africa in SON, the correlation between $-v'$ and $T'$ becomes significantly
more positive, which may, in combination with the amplified temperature gradient, contribute to increasing South African $T_{\mathrm{VAR}}$ through enhanced generation efficiency (see right column of Fig. 8 in SON) and subsequent downstream advection.

The role of $F_{\mathrm{horiz}}$ is further assessed by investigating how it affects biases in $T_{\mathrm{VAR}}$ among the reanalyses. It is achieved here by correlating the trends in $T_{\mathrm{VAR}}$ averaged over a reference region, assessed independently for each reanalysis, with trends in $F_y^{\mathrm{eff}}$ at each grid point (heterogeneous corre-
lation; Fig. 10). The correlation is evaluated in the reanalysis dataset space, indicating the relationship between reference $T_{\mathrm{VAR}}$ and $F_y^{\mathrm{eff}}$ trend biases among reanalyses. Since the correlation is assessed for each grid point, a map showing the relationship between $F_y^{\mathrm{eff}}$ trends and reference $T_{\mathrm{VAR}}$ trend
is obtained. The use of such a map is motivated by the fact that remotely generated $T_{\mathrm{VAR}}$ by $F_{\mathrm{horiz}}$ may affect the reference region through horizontal advection of $T_{\mathrm{VAR}}$ by the basic-state circulation. The same analysis is repeated for the three regions of interest (panels in Fig. 10). An assessment of
the spatial extent of $T_{\mathrm{VAR}}$ trend biases is also performed by correlating $T_{\mathrm{VAR}}$ trends at each grid point with the reference $T_{\mathrm{VAR}}$ trend (homogeneous correlation; contours in Fig. 10).

We find from the homogeneous correlation map that $T_{\mathrm{VAR}}$ trend biases in SON over South Africa (Fig. 10, first row) are not geographically confined but tend to accompany, as indi-
60 cated by large areas of positive correlation, biases of the same sign around 30° S at almost all longitudes. Similarly, we also observe from the heterogeneous correlation a generally positive association with $F_y^{\mathrm{eff}}$ trends at a similar latitude band. In other words, biases affecting South Africa tend to be part
of SH-wide biases at similar latitudes. The biases affecting $T_{\mathrm{VAR}}$ trends in DJF around eastern South America (Fig. 10, second row) are more geographically confined in comparison with a more modest correlation with $T_{\mathrm{VAR}}$ trends (homogeneous correlation) over other SH sectors as well as positive
correlations with $F_y^{\mathrm{eff}}$ trends (heterogeneous correlation) that are more concentrated near South America. Finally, $T_{\mathrm{VAR}}$ trend biases in SON over southern Australia (Fig. 10, third row) tend to be associated with $T_{\mathrm{VAR}}$ trend biases (homogeneous correlation) of the same sign in midlatitudes $\sim 40$–
55° S over the south Pacific, Atlantic, and Indian oceans, and those of the opposite sign over the subtropics. Concerning the relationship with $F_y^{\mathrm{eff}}$ (heterogeneous correlation), there is notably a covariability with $F_y^{\mathrm{eff}}$ biases around South America. These findings indicate that biases in $T_{\mathrm{VAR}}$ trends in re-
analyses are not locally confined. Instead, they are part of broad biases in mean-state trends and their interactions with subweekly variability.

## 4   Discussion and conclusions

In summary, reanalysis datasets generally agree well concerning the climatological features (1980–2010) of $T_{\mathrm{VAR}}$ in
the SH (Fig. 3). It is maximized in the south Atlantic and Indian oceans. Local maxima are also observed near or over landmasses, specifically in SON and DJF over southern Australia, throughout the year around South Africa, and in JJA
and SON around Argentina, indicating an anchoring of subweekly variability by land–sea thermal contrasts (Fig. 1). $T_{\mathrm{VAR}}$ is primarily generated through horizontal advection ($F_{\mathrm{horiz}}$) and offset by vertical motion ($F_{\mathrm{vert}}$) (Fig. 2). The spatial patterns of $F_{\mathrm{horiz}}$ and its seasonality mirror that of
$T_{\mathrm{VAR}}$ with, for instance, maxima over South Africa and Australia in SON and South America in JJA and SON. Among all datasets considered, NCEP-NCAR (R1) and NCEP-DOE (R2) show noticeable negative biases around the midlatitude $T_{\mathrm{VAR}}$ maximum that is associated with the storm track over
the ocean (Figs. 3 and 4). This finding is in agreement with the substantial reduction of eddy APE identified in NCEP-DOE (R2) (Sang et al., 2022), which is attributed to its coarser model resolution. Over SH landmasses, however, the biases are greatly reduced, which may be due to the greater
availability of observations. It is noted by NOAA's Physical Sciences Laboratory that NCEP-NCAR (R1) is affected by

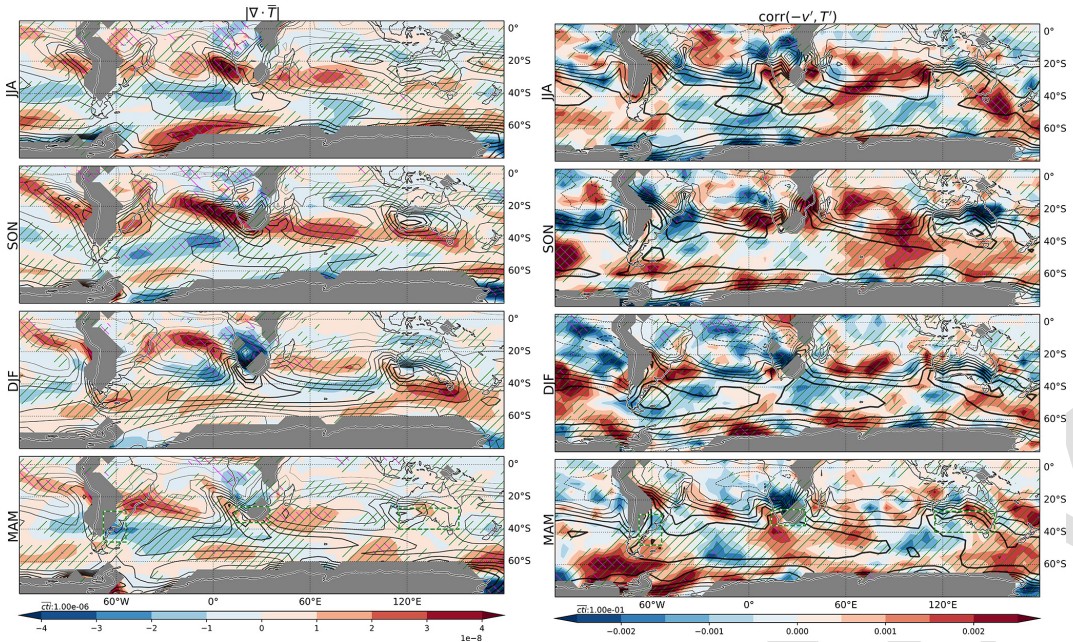

**Figure 9.** Same as in Fig. 8, but for (left) $\left|\nabla\overline{T}\right|$ (K m$^{-1}$ yr$^{-1}$) and (right) the correlation between $-v'$ and $T'$.

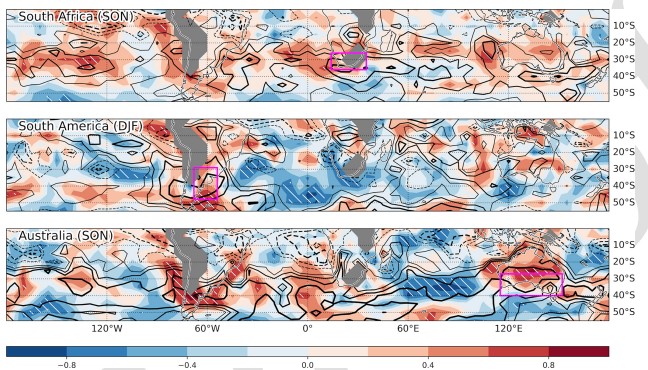

**Figure 10.** Sources of inter-reanalysis bias evaluated by correlating among reanalyses trends in $F_y^{\mathrm{eff}}$ at each grid point with trends (1980–2010) in $T_{\mathrm{VAR}}$ (shading; heterogeneous correlation) averaged over three representative regions as indicated in individual panels with purple rectangles. Significant correlations ($p < 0.05$) are indicated with white hatching. Note that the season, which is also indicated in each panel, differs among the regions. For reference, the correlation is also assessed for $T_{\mathrm{VAR}}$ trends at each grid point (homogeneous correlation; black contours; 0.2 intervals; solid and dashed lines for positive and negative correlations, respectively; the 0 lines are omitted).

the assimilation of erroneous surface pressure data in the SH. This error was subsequently corrected in NCEP-DOE (R2), thus it is not the cause of the important biases observed in both datasets. The use of these two older datasets is generally discouraged by the SPARC Reanalysis Intercomparison Project (S-RIP) (Fujiwara et al., 2022).

We find a good agreement concerning the significant positive $T_{\mathrm{VAR}}$ trends (1980–2022) affecting South America in DJF and South Africa in SON (Fig. 5). Although most of the reanalyses agree concerning positive trends over southern Australia in SON, they are not statistically significant for the satellite era (1980–2022). The latter trends are, however, significant when considering a longer period (1954–2022) provided by some of the datasets (Fig. 7), most likely due to the larger sample size, and for the most recent decades when the amplification of $T_{\mathrm{VAR}}$ has accelerated. These trends are also observed in gridded, station-based temperature records, indicating that they are not the result of discontinuities in data assimilation. These three sectors sometimes exhibit discontinuities in $T_{\mathrm{VAR}}$ trends. For instance, $T_{\mathrm{VAR}}$ in SON over South America tends to amplify before the satellite era but decreases afterward (Fig. 7). We observe similar discontinuities in trends surrounding the beginning of the satellite era in surface observations and reanalyses, indicating that these are not the result of discontinuities introduced by the advent of the assimilation of satellite observations. They are more likely due to multidecadal variability. This is also supported by the fact that surface-input reanalyses, whose assimilated observations are more constant over the period considered, also capture similar modulations in the trends.

Our results are consistent with the column-integrated SH-wide increases in wintertime eddy KE and moist static energy fluxes observed during 1979–2018 in reanalyses (Chemke et al., 2022). They are, however, less consistent with the intensification and poleward shift of the summertime (DJF) polar-front jet. This is observed since the beginning of the satellite era as a result of the stratospheric ozone

depletion (Orr et al., 2021), although pausing since 2000 due to a hint of its recovery (Banerjee et al., 2020). From these changes, one would expect a weakening of temperature variability over South America. Perhaps this indicates that merid-

5 ional shifts in the jet stream and associated changes in eddy KE are not necessarily good indicators for near-surface temperature variance. It is worth noting that the prominent spatial inhomogeneities observed in $T_{\text{VAR}}$ trends suggest that it is necessary to avoid using large-scale spatial averaging, such

as the zonal mean, when interested in the potential socioeconomic impacts of changing atmospheric variability.

Overall, the spatial patterns of $F_{\text{horiz}}$ trends and their efficiency are similar to those of $T_{\text{VAR}}$ trends, indicating that eddy fluxes of heat against the seasonal-mean gradient of

15 temperature are the prime driver of amplified subweekly temperature variance. Whereas over South Africa and Australia it is concomitant with a local amplification of the meridional temperature gradient that is more prominent in SON, it is ascribed primarily to a change in the structure of subweekly

eddies over South America in DJF that enhances their efficiency in transporting heat across the seasonal-mean temperature gradient. While the former can be deduced simply from large-scale temperature trends, the latter requires more detailed knowledge of how eddies react to seasonal-mean flow

changes and cannot be inferred from future trends in temperature gradients alone.

One potential source of bias in $T_{\text{VAR}}$ and $F_{\text{horiz}}$ trends among reanalyses is the impact of the representation of sea surface temperature (SST) on the development of atmo-

30 spheric eddies. Masunaga et al. (2018) showed that a version of JRA-55C with improved SST resolution, JRA-55CHS, better represents mesoscale atmospheric structures up to the mid-troposphere. Many of the reanalysis products considered transitioned through different SST datasets throughout

their integration period (Table 4 of Fujiwara et al., 2017) and these discontinuities could have introduced changes in $T_{\text{VAR}}$. It is, however, challenging to assess the impact of SST representation in the context of this comprehensive comparison of reanalyses because of a lack of controlled experiments.

We found, however, a tendency for datasets with amplified SST trends in the SH to also show amplified $T_{\text{VAR}}$ trends (Fig. 11). For instance, we find evidence that reanalyses with more pronounced SST trends in the subtropical Pacific and Indian oceans tend to have greater $T_{\text{VAR}}$ trends over South

Africa. This simple analysis, however, does not account for SST resolution and suffers from a small sample size (five reanalyses), with strong influence from NCEP-NCAR (R1) as an outlier. Further confirmation of the role of SST is required in future work by carefully considering the transitions in the

assimilation of various SST products.

Concerning the value of surface-input reanalyses (20CRv2c, 20CRv2, and ERA-20C), we have found that they capture relatively well both the climatology and trends in $T_{\text{VAR}}$ despite the limited observations being

assimilated. In fact, their representation of $T_{\text{VAR}}$ is similar

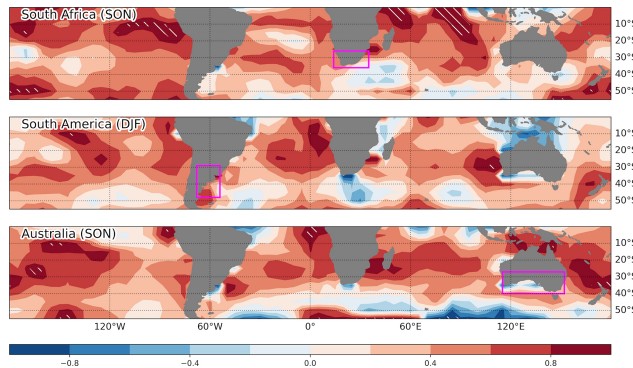

**Figure 11.** Same as in Fig. 10, but for SST trends (1980–2010; heterogeneous correlation; shading) based on a subset of reanalyses (ERA5, ERA-Interim, JRA-55, MERRA-2, NCEP-NCAR).

to or sometimes even better than that of NCEP-NCAR (R1) and NCEP-DOE (R2), which benefit from full data assimilation over the 1979–2022 period. This suggests that they could potentially be used to reliably assess long-term changes in $T_{\text{VAR}}$ over the past century, either due to external 60 forcing or multidecadal internal variability. Similarly, the conventional-input JRA-55C, which does not assimilate satellite observations, also agrees well with other reanalyses, indicating that satellite observations are not absolutely necessary to constrain $T_{\text{VAR}}$ near the surface over the sectors 65 studied here.

It is important to mention that by comparing seasonally averaged $T_{\text{VAR}}$ and generation/dissipation terms among the reanalyses, we are assessing their statistical representation of subweekly variability, not their ability to capture specific 70 weather events. Observations in some sectors may sometimes insufficiently resolve migratory weather systems so that the model component of reanalyses is primarily responsible for generating dynamical variability. This model dependence may be especially important in surface-input reanal- 75 yses over vast oceanic sectors. In ensemble-based reanalyses, such as 20CR, this could contribute to suppressing a part of internal variability that is not properly constrained by observations. Assessing the ability of reanalysis datasets to adequately capture subweekly variability in a deterministic 80 sense, i.e., capturing the occurrence of specific events, will be the topic of future work.

*Code availability.* Code can be provided upon request.

*Data availability.* JRA-55 (Japan Meteorological Agency, 2013), JRA-55C (Japan Meteorological Agency, 2015), CFSR (Saha et 85 al., 2010b), CFSv2 (Saha et al., 2011), and ERA-20C (European Centre for Medium-Range Weather Forecasts, 2014) were obtained from the UCAR research data archive. MERRA (Global Modeling and Assimilation Office, 2008) and MERRA-2 (Global Mod-

eling and Assimilation Office, 2015) were obtained from the NASA Goddard Earth Sciences Data and Information Services Center. ERA5 (Hersbach et al., 2023) was obtained from the Copernicus climate data store. ERA-Interim (ECMWF, 2009) was obtained from the ECMWF data server (now decommissioned). NCEP-NCAR (R1) (National Centers for Environmental Prediction, 1994), NCEP-DOE (R2) (National Centers for Environmental Prediction, 2000), 20CRv2c (NOAA's Physical Sciences Laboratory, 2015), and 20CRv3 (NOAA's Physical Sciences Laboratory, 2019) were obtained from NOAA's Physical Sciences Laboratory. The URL/DOI of each dataset and the last access date are indicated in the references.

*Supplement.* The supplement related to this article is available online at: https://doi.org/10.5194/wcd-6-1-2024-supplement.

*Author contributions.* PM led and coordinated the various components of the study throughout. All authors (PM, SKB, MN, HN, YK) discussed the results and aided in their interpretation. PM took the lead in writing the manuscript.

*Competing interests.* The contact author has declared that none of the authors has any competing interests.

ther geographical representation in this paper. While Copernicus Publications makes every effort to include appropriate place names, the final responsibility lies with the authors.

*Special issue statement.* This article is part of the special issue "The SPARC Reanalysis Intercomparison Project (S-RIP) Phase 2 (ACP/WCD inter-journal SI)". It is not associated with a conference.

*Acknowledgements.* The authors sincerely thank the two anonymous reviewers and the editor for giving us insightful and constructive comments.

*Financial support.* This research has been supported by the Japan Society for the Promotion of Science (grant nos. JP19H05702, JP19H15703, JP22H01292, and JP23H01241), the Japan Science and Technology Agency, Co-creation place formation support program (grant no. JPMJPF2013), Japanese Ministry of the Environment (grant no. JPMEERF20222002), and the Ministry of Education, Culture, Sports, Science and Technology (grant no. JP-MXD0722680395, JPMXD1420318865, and ArCS-II).

*Review statement.* This paper was edited by Tim Woollings and reviewed by two anonymous referees.

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

**Remarks from the typesetter**

TS1    Please confirm updated equation.

TS2    For this change we need the approval from the editor. Please write a statement, why this change need to be made and we will send it to the handling editor. Thank you.

TS3    Please confirm updated following expression.