# Peer review of "A rise in subweekly temperature variability over Southern Hemisphere landmasses detected in multiple reanalyses"

_EGUsphere, 2023_

## Author Response (AR1)

**Responses to the reviewers**

We would like to thank the two reviewers for their constructive comments and remarks. Our responses are written below in blue. We note that on Fig. 11, we have removed the homogeneous correlation with $T_{VAR}$ as it was already shown in Fig. 10. We also indicate that NCEP-NCAR (R1), as an outlier, strongly contributes to the correlations shown in Fig. 11 [line 404]. Line numbers indicated correspond to the tracked-changed version.

**Reviewer 1:**

The article explores the agreement between datasets in depicting subweekly variability in the Southern Hemisphere to then use the reanalysis datasets to explore the trend of this variability. Overall there is a good agreement between reanalysis, specially between the new generation (ERA5, MERRA2, etc). The authors find some positive trends in subweekly temperature variability although it depends on the season considered and the region. In addition, this trend in not always significant.The topic fits the scope of this journal and the manuscript is generally well-written. Therefore, the manuscript is valuable to be published after some changes.

Major comment: I understand that nowadays research articles have titles that are eye-catching and also highlights the main results, instead of summarizing the type of research conducted. However, from the analysis of Figure 5 I don't think that the title corresponds well with the results. The authors find some positive significant trends in some regions of the SH landmasses (mainly midlatitudes) but this do not occur in all seasons (actually it is only in midlatitude South America in DJF and MAM and South Africa in SON and southern Australia in JJA). So I would like to ask the authors to change the title accordingly. Actually, considering that this article is part of a special issue on reanalysis I find more interesting the good agreement between reanalysis over the Southern Hemisphere in representing trends.

Thank you for the useful comment. We have changed the title to the following which is more representative of our results: "Seasonally dependent rise in subweekly temperature variability over Southern Hemisphere landmasses detected in multiple reanalyses".

Methods:

I'm assuming that the authors use daily reanalysis data. The different reanalysis used in this study also have different temporal resolution (some of them has 6-hourly data, hourly data available, etc). However I don't find in the document any reference to this. Could the authors briefly explain how they treat the differences in the temporal resolution between reanalysis as they did with the spatial resolution? Also a brief discussion on how these differences may impact the results would be appreciated.

We confirm that we have used daily-mean data to conduct our analyses. Daily means were calculated from four time steps that are common to all reanalysis datasets (0, 6, 12, and 18 UTC). We add this information to the "Reanalysis data" section of the revised manuscript. Since we used only common time steps for our analysis, differences in the temporal resolution of data provided by the reanalysis centers do not impact our study.

Results:

Line 21: Talking about extratropical variability and immediately after mentioning tropical storms does not sound coherent to me. I'd use mesoscale storms or something similar

What we referred to here are the hurricanes and typhoons propagating poleward and undergoing extratropical transition causing severe weather in the mid-latitudes. We have clarified this by writing "tropical cyclones migrating poleward" instead and inserting it after "midlatitude cyclones/anticyclones". Thank you for the suggestion, we have also added mesoscale storms to the list.

134: I'd remove the expression between commas "like the Antarctic polar frontal zone" as the Antarctic Polar frontal zone does not entirely owe its existence to L-S contrasts. Removing this won't change the meaning of the sentence.

We agree that this sentence can be confusing. We have moved the expression to the end of the sentence which will prevent the reader from interpreting the Antarctic polar frontal zone as being a land-sea contrast. What we mean here is the following: Their [the secondary maxima] presence indicates that land-sea contrasts have the potential to anchor subweekly variability, like the Antarctic polar frontal zone [can anchor subweekly eddies].

148-149: The relationship between the local maxima and stationary waves in the SH needs a reference. Otherwise, it should be removed. To my knowledge the wave 1 is the QS wave that dominates the variability in the Southern Hemisphere (see for instance Quintanar and Mechoso 1995).

We have added the following reference [line 155] which shows stationary waves in the geopotential and temperature fields: Wallace, J. M.: The climatological mean stationary waves: observational evidence, in Large-Scale Dynamical Processes in the Atmosphere, edited by B. J. Hoskins and R. P. Pearce, pp. 27–53, Academic Press., 1983.

Near the surface, anomalies as departures from the zonal mean do not show the signature of a pure wavenumber-1 pattern (Figs. 2.15 &. 2.16). It most likely reflects an important contribution from wavenumber-2 and 3.

Figure 3: I find the selection of SON a bit arbitrary. Can the authors briefly discuss other seasons and include the corresponding figures as supplementary material? Another option could be showing the biases only for reanalysis included in the REM in the main document, and put the remaining reanalysis in the supplementary material. There is also a label "cti: 1.00+e00" next to the bar that does not make any sense to me

Thank you for the comment. We have added other seasons in supplementary material to briefly discuss them [lines 187-189]. Essentially, the large-scale features of these biases tend to be similar in other seasons. The "cti" label indicates the contour interval of the climatology. We will clarify this in the revised manuscript.

Line 223: What do you mean by "clearer"? Are the trends higher or lower?

We meant that the signal-to-noise ratio is higher (similar noise but a larger trend). We rewrite this section to be more clear: "… surface TVAR trends have a greater signal-to-noise ratio than the…"

Line 304-306: I don't understand how do you correlate the Tvar trend of each reanalysis (one value per reanalysis) against the reference

The "reference" is assessed independently for each reanalysis, thus allowing to evaluate correlations in the reanalysis space. We have amended this section to improve the clarity [lines 310-311].

Conclusions:

Line 356: Chemke et al 2022 only refers to CMIP6 data. Are the authors sure that the comparison with CMIP5 data cited in the article comes from Chemke 2022? I could not find it. Nevertheless, I don't find the reference to CMIP6/5 data useful at all since Chemke already pointed out that models do not represent the observed trend well and therefore might underestimate the future trend. I'd remove the reference to CMIP data there. The authors can still speculate on the agreement between future changes in EKE and low level temperature variability.

While EKE trends are shown for CMIP5 in [1], they are shown for CMIP6 models in [2].

[1] Chemke, R.: The future poleward shift of Southern Hemisphere summer mid-latitude storm tracks stems from ocean coupling, Nat. Commun., 13(1), 1–9, doi:10.1038/s41467-022-29392-4, 2022

[2] Chemke, R., Ming, Y. and Yuval, J.: The intensification of winter mid-latitude storm tracks in the Southern Hemisphere, Nat. Clim. Chang., 12(6), 553–557, doi:10.1038/s41558-022-01368-8, 2022)

We agree with the reviewer that this part of the discussion could be removed. We only keep the aspects related to observation-based trends in the revised version.

Line 345-346: I would not compare directly reanalysis data at 850hPa to surface station based data. It could be that all reanalysis have problems representing the trends at 850hPa and the conclusions of sfc processes amplifying the trends can't be drawn from the comparison you just made (I agree with you that sfc processes may amplify trends but I can't arrive to this conclusion from the results you have shown)

Thank you for the comment. We have removed the speculation that surface processes amplify the trends. We believe that it is nonetheless useful to compare reanalyses at 850 hPa with observations at the surface as this allows us to assess if long-term variability and breakpoints in the trends are artifacts or likely real features.

**Reviewer 2:**

In this study, the authors assess the climatology and interannual variability of subweekly temperature variance among multiple reanalyses. Results show that there is a good agreement for the climatological temperature variance and dominant sources and sinks of variance. The authors also point out that there is a good agreement for the positive trends in subweekly variability over South Africa and South America. The analyses are clear, the results are reliable, and the writing is good. I have some concerns and suggestions for the authors to consider for improving their manuscript.

Comments:

What are the reasons for the largest bias of NCEP-NCAR (R1) and NCEP-DOE (R2) that are modern full-input datasets from the REM climatology? How the observarions of R1 and R2 assimilation are distributed? And how the assimilated obsevations can affect the subweekly variability and generation term Fhoriz?

Thank you very much for the comment. In lines 53 and 335, we now refer to Sang et al. (2022), who noted that lower-resolution products under-represented baroclinicity, leading to weaker variance.

It is debatable that NCEP-NCAR and NCEP-DOE are modern reanalyses. We now indicate that their use is discouraged in the S-RIP report [1], and that NCEP-NCAR was affected by the assimilation of erroneous surface pressure data in the SH as indicated by NOAA's PSD (https://psl.noaa.gov/data/reanalysis/problems.shtml) [lines 349-354].

Maps of observation density are typically not provided by reanalysis centers in their reference publications. The scarcity of observations in the SH compared to the NH, especially over the ocean, is nicely illustrated in [2] which we now refer to [line 50].

The impact of data assimilation would have to be quantified by evaluating the analysis increment, i.e., the difference between the forecast and the final state after data assimilation. This increment is generally not provided by reanalysis centers, nor is the forecast. For instance, they are not provided for NCEP-NCAR and NCEP-DOE. This analysis could be assessed for some of the reanalyses that do provide forecasts, but it does not fit within the scope of this paper, which is to carry out a comprehensive intercomparison of reanalyses. It should be carried out in future work.

[1] Fujiwara, M., Manney, G. L., Gray, L. J. and Wright, J. S.: SPARC Reanalysis Intercomparison Project (S-RIP) Final Report, SPARC Repo., SPARC, 2022., 2022.

[2] Noone, S., Atkinson, C., Berry, D. I., Dunn, R. J. H., Freeman, E., Perez Gonzalez, I., Kennedy, J. J., Kent, E. C., Kettle, A., McNeill, S., Menne, M., Stephens, A., Thorne, P. W., Tucker, W., Voces, C. and Willett, K. M.: Progress towards a holistic land and marine surface meteorological database and a call for additional contributions, Geosci. Data J., 8(2), 103–120, doi:10.1002/gdj3.109, 2021.

The Eq. (1) explains the tendency of TVAR, not the climatology or trend of TVAR. For equilibrium, the tendency of TVARis close to 0 and Fhoriz and Fvert always cancel each other out. The characteristics of climatolgical subweekly variability may not simply be explained by the source or sink terms the right-hand side of Eq. (1). Please integrate the right-hand side to obtain the source for TVAR or differentiate the left side to obtain the tendency of TVAR.

No equation can fully explain the climatology of a field. One can only assess the balanced processes (generation and dissipation) leading to this climatology. To this end, we have assessed the climatology, i.e., averaged over time the generation/dissipation terms (Figs. 2 and 4), which is proportional by a factor L (length of period averaged) to the suggestion of the reviewer to integrate the right-hand side terms.

Concerning trends, a common approach, the one adopted here, is to assess trends in the leading source terms [e.g., 1, 2]. Integrating the generation/dissipation terms, or right-hand-side terms, as suggested by the reviewer would prevent us from understanding how these forcings change over time. The left-hand side of Eq. (1) already expresses $T_{VAR}$ tendency, taking another derivative would not be helpful to assess

trends. We note that to assess sources or sinks, one needs to look at the tendency equation as we have done. It does not require integration.

[1] Orr, A., Lu, H., Martineau, P., Gerber, E., Marshall, G. and Bracegirdle, T.: Is our dynamical understanding of the circulation changes associated with the Antarctic ozone hole sensitive to the choice of reanalysis dataset?, Atmos. Chem. Phys., 1–35, doi:10.5194/acp-2020-1288, 2021.

[2] Chen, G. and Held, I. M.: Phase speed spectra and the recent poleward shift of Southern Hemisphere surface westerlies, Geophys. Res. Lett., 34(21), L21805, doi:10.1029/2007GL031200, 2007.

What is the time-scale of the trends of TVARinvestigated in the manuscript? Whether the effects of global warming are included? What internal or external forcings can explain the trends. That is how can we understand the trend of efficiency term?

As indicated in figure captions (e.g., Fig. 5), most figures show trends over 1980-2022, but we have also considered shorter and longer trends as shown in Fig. 7 and described in the associated discussion. We indicated for instance at line 248 that: "The South-American trends are, however, positive when assessed for the ~1954-1980 period. Assessing trends over such short periods, however, may capture inter-decadal variability not associated with climate change or discontinuities in assimilated observations, such as the beginning of satellite assimilation in 1979 in full-input datasets providing data before the satellite era....".

We indicate at line 398 that the trends could be due to external forcing but also due to multidecadal variability. We have added "multidecadal internal variability" to clearly indicate that it is not the result of climate change [line 417]. We wish to avoid speculating on the causes without concrete evidence. Further analyses on the causes of multidecadal variability should be the topic of future work.

The manuscript focuses on the climatology and trend of different seasons of three landmasses in the Southern Hemisphere, which are mainly descriptive. Please give more conclusions about seasonal differences from the perspective of physical mechanisms, which may be more impressive.

Thank you for the comment. Figures 8-10 and their associated discussions all aim to shed light on the physical processes explaining the trends for different seasons. At line 369 of the conclusion, we summarize our findings on the role of $F_{horiz}$ in the trends. It is true however that we could more explicitly mention the role of the seasonal cycle of $F_{horiz}$ in the seasonal cycle of $T_{VAR}$. To this end we have improved our discussion of the results in the revised manuscript [line 153]: "Other maxima in Fhoriz and this gradient found over eastern South America, South Africa, and southern Australia exhibit the same seasonality as TVAR, i.e., peaking in SON over South Africa and Australia and affecting a larger fraction of South American landmass in JJA and SON". We have also added to our conclusion at line 344 that: "The spatial pattern of $F_{horiz}$ and its seasonality mirrors that of $T_{VAR}$ with, for instance, maxima over South Africa and Australia in SON and South America in JJA and SON"

---

## Author Response (AR2)

We thank the editor for carefully checking the manuscript and his invaluable suggestion to enhance the title. We agree with the proposed change and have subsequently made the necessary modifications to the manuscript files.